# Carbon nitride caught in the act of artificial photosynthesis

**Daniel Cruz** [1,4], **Sonia Żółtowska** [2,4], **Oleksandr Savateev** [2,3], **Markus Antonietti** [2] **& Paolo Giusto** [2] ✉

Covalent semiconductors of the carbon nitride family are among the most promising systems to realize "artificial photosynthesis", that is exploiting synthetic materials which use sunlight as an energy source to split water into its elements or converting $CO_2$ into added value chemicals. However, the role of surface interactions and electronic properties on the reaction mechanism remain still elusive. Here, we use in-situ spectroscopic techniques that enable monitoring surface interactions in carbon nitride under artificial photosynthetic conditions. We show that the water adsorption and light illumination cause changes of the surface electron density, which activate the photocatalyst and enable the water splitting process. Our results reveal critical details on the photocatalytic mechanism, which proceeds through proton-coupled electron transfer, and provide key information to design more efficient photocatalyst for artificial photosynthesis.

Since the discovery of the ability of carbon nitride to act as a photocatalyst for splitting water into its elements[1], tenth of thousands of works have been published reporting continuous improvements of the system including structural changes, doping, defect design, co-condensation, or heterojunction formation[2,3]. Carbon nitrides meanwhile have long expanded the operation range of natural photosynthesis: they can generate hydrogen with apparent quantum yields exceeding 60%[4–6]. Furthermore, this class of materials has been successfully applied to several impelling challenges in catalysis: they can selectively mono-oxidize methane to methanol[7], or they can accomplish so-called deep oxidations well below their standard potential, such as the oxidation of chloride salts to $Cl^+$[8]. Despite the enormous progress in the field of transient spectroscopies[9,10] and molecular simulations[11–13] supporting the improvements and our molecular understanding, we still know little about the fundamental steps controlling the photochemical process of water splitting.

X-ray photoelectron spectroscopy (XPS) and near-edge X-Ray absorption spectroscopy (NEXAFS) are surface sensitive spectroscopic techniques that enable the determination of the electron density of most elements close to their nuclei. Recently, in-situ irradiated XPS was used to monitor the migration of photoexcited electrons in a hybrid photocatalyst and to correlate the elemental peaks shift to the electronic changes upon light irradiation[14]. Since the binding environment with atoms of different electronegativity, conjugation, ligation, influences the local electron density, but also collective electron properties, such as the Fermi level, science finds in these techniques powerful tools to carefully determine local electron density in the form, for example, of partial charges on specific atoms in a localized binding environment. The development of synchrotron radiation and advanced spectroscopy techniques based on differentially pumped apertures close to the surface sample[15] enable nowadays to conduct spectroscopic experiments at near ambient pressures (thereby in restricted presence of educts and products) to mimic closely realistic reaction conditions of volatile substrates and products, such as water, hydrogen, and oxygen[16,17]. Scientists could thereby determine if and how water binds to the catalyst at the interface and how light irradiation changes the electronic structure throughout photon absorption processes, which is a previously untapped physical response of the so-called "excited state". For instance, it will be shown that the water adsorption step plays a pivotal role in the photocatalytic water splitting by altering the surface electronic

[1]Fritz Haber Institute of the Max Planck Society, Department of Inorganic Chemistry, Faradayweg 4-6D, 14195 Berlin, Germany. [2]Max Planck Institute of Colloids and Interfaces, Colloid Chemistry Department, Am Mühlenberg 1, 14476 Potsdam, Germany. [3]Present address: Department of Chemistry, The Chinese University of Hong Kong, Shatin, New Territories, Hong Kong, China. [4]These authors contributed equally: Daniel Cruz, Sonia Żółtowska. ✉e-mail: Paolo.Giusto@mpikg.mpg.de

properties of the photoactive carbon nitride semiconductor, creating an activated state through pre-polarization.

Here, the combination of in-situ spectroscopic techniques, such as XPS and NEXAFS, coupled with a time of flight mass spectrometry (TOF-MS) detector at near ambient pressure enables the tracking the changes occurring during the photocatalytic process. In the following, we show that the adsorption of heavy water molecules on the surface of the carbon nitride occurs via the formation of hydrogen bonds, where the carbon nitride acts as donor of electron density. Upon light irradiation, the electronic excitation of the catalysts promotes the stretching and breaking of the O-D bonds, resulting in the formation of deuterium and oxygen as final products. With these experiments, it was possible to capture the stable surface states of the carbon nitride at a chemical level during the photocatalytic water splitting process. From a practical point of view, these findings are crucial for improving the design of heterogeneous catalysts for artificial photosynthesis.

## Results

### Adsorption of heavy water

The experiments were pursued using a thin film of polymeric carbon nitride synthesized by means of chemical vapor deposition (Fig. S1)[18]. The film is well-known to be photochemically active, as previously shown in batch and flow reactors for organic photochemistry[19]. The XPS and NEXAFS characterization of the bare sample reveals the typical features of carbon nitride thin films (Fig. S2A–E)[18,20,21]. Furthermore, the valence band XPS (VB-XPS) (Fig. S2E) spectrum shows that the highest occupied molecular orbital (HOMO) energy level of the carbon nitride thin film is +1.59 eV, in good agreement with previously reported values[18]. It is worth noticing the absence of any gold signal of the substrate that enables us to rule out its contribution to the photocatalytic process (Fig. S3).

In-situ near ambient pressure (NAP)-XPS and -NEXAFS experiments were designed on the 1.5 GeV storage ring at MAX IV Laboratory in Lund, Sweden (details of the experimental setup and conditions are available in Supplementary Materials and pictures in Fig. S4). We explored in detail the effect at 0.2 mbar $D_2O$ vapor pressure, as it is expected to provide the thinnest adsorption layer while causing significant changes in the XPS and NEXAFS spectra. The adsorption binding at such low pressures is not trivial and relies on a rather high binding enthalpy to enrich the carbon nitride surface with the water molecules diluted in the gas phase. Recent modeling data indeed revealed very high adsorption enthalpies ($\Delta H \sim$ -95 kJ/mol) between water and an ideal carbon nitride structure[22]. Such high adsorption energies point to highly favorable interactions between the water molecules and the carbon nitride, securing a rather complete binding of water at the adsorption sites under experimental conditions.

Unexpectedly, the introduction of $D_2O$ into the system changes significantly the electronic structure of the carbon nitride (Fig. 1). The adsorption of water molecules is manifested in a shift of the XPS peaks' maxima towards higher binding energies, especially of the nitrogen, as the latter plays the role of a potential electron donor to the heavy water (Fig. 1A). This behavior lowers the whole electron density of the surface of the semiconductor, manifested as a shift of the spectroscopic features also for the carbon atoms (Fig. 1B). Indeed, the shift is particularly pronounced for the catalyst's edge functionalities (-C-N = C-) pointing towards a preferential adsorption at these sites, as confirmed also by the NEXAFS spectra (Fig. 1C, D). We can only speculate at this moment an oxygen back-binding to the carbon in the structure which stabilizes the overall hydration structure of carbon nitride photocatalyst[11]. As a consequence of the donation of electron density into hydrogen bonds with water, the valence band position of the carbon nitride film shifts by 1.04 eV to higher binding energies, as clearly visible in the VB-XPS spectrum (Fig. 1E). This points to a localization of the otherwise collective semiconductor valence band electrons for the formation of hydrogen bonds, where the corresponding charge density is

transferred from the carbon nitride to the heavy water (Fig. 1F). In simple terms, we may say that the water participates in the formation of a joint hybrid semiconductor with the carbon nitride, resulting in a different electronic structure.

The quantified changes in collective electron density (as integrated over in the valence band spectrum and position at low binding energies) represents a more favorable starting point for the photocatalytic process. While electron density is shifted to the water, the electron depleted carbon nitride exhibits a significantly higher oxidation potential, thus remarkably simplifying the otherwise more difficult step of water oxidation[23–26]. This band shift was, to the best of our knowledge, never observed before and it is the first step to activate artificial photosynthesis. Therefore, we convey that the reaction environment plays a critical role in determining the real oxidation potential of carbon nitrides (Supplementary Notes 1, 2). Potentially this is also true for other semiconductors and catalysts, especially when strong catalyst-substrate interactions, such as hydrogen bonds, occur.

### Illumination with solar simulator

Upon light illumination, the carbon nitride absorbs photons and excites electrons from the valence band to the conduction band, most probably localizing them at the water adsorption sites as they are the most electron deficient sites. This added photoelectron density indeed is seen in the XPS spectra of the illuminated species and accounts for a shift of the N- and C-peaks towards lower binding energies (Fig. 2A, B). The shift is remarkable as it is visible in the overall sample structure even on the timescale of the XPS experiments. Usually, photoexcited states have a rather short lifetime of a few nanoseconds, too short to be detected by XPS[9]. The fact that the experimental effect is visible on this timescale means that the charge density is already stabilized in a long-living state. The shift towards lower binding energies, for the nitrogen and carbon features, points to the formation of rather stable N-photoelectron $\cdots D^+$ and N $\cdots$ D-O$^\cdot\cdots$ C species, which are electronically stabilized by conjugation over the carbon nitride structure, as reflected by the concomitant shift of the whole spectrum. On the other hand, we observe only negligible variations in the intensity of the NEXAFS spectra (Fig. 2C, D), where the only significant change occurs in a restricted region of the C K-edge (green area, 286-287.1 eV), conveying the pivotal contribution of this carbon site to this step. These changes in the electronic environment are accompanied by a minor shift of the VB-XPS spectrum towards lower binding energies, accounting for a slightly lower oxidation potential. This is meaningful, as the delocalized extra photoelectrons will partly lower the electrophilicity of the joint system, i.e. if all the atomic sites are more electron rich also the collective state has to be more electron rich (Fig. 2E).

We interpret the above as the spectroscopic reflection of a rather stable intermediate structure supporting the proton-coupled electron transfer mechanism (PCET)[27–29], where the electrons are waiting delocalized along the conjugated reaction site Coulombically-stabilized by a proton. The corresponding hole arising from the photochemical excitation, at this stage, has already vanished from the conjugated system to create, with the missing DO$^-$, a localized DO$^\cdot$ surface specie (which is expected to recombine later to $D_2O_2$, Supplementary note 3)[11,30–33]. Notably, this parallel process is not recorded in the spectroscopic features of the photocatalyst since the hole transfer to the heavy water phase has already been accomplished on the experimental timescale of the spectroscopic measurements (Fig. 2F). Note that such a surface bound radical is usually high in oxidation strength (about +2.0 V)[34], a level that chemically only the adsorption-activated semiconductor is able to generate. This is indeed photosynthesis caught-in-the-act. We are only missing the kinetically faster hole transfer, presumably from a carbon site directly to the rebound the oxygen of the heavy water.

The significant shift of the XPS peaks of most structural units upon exposure to $D_2O$ first towards higher binding energies and, upon light irradiation, to lower binding energies points towards a process

cascade where binding precedes reaction. Here, the carbon nitride first chemically donates electron density to the heavy water via hydrogen bonding, while the light illumination at the next step, causes the photocatalyst to be formally reduced, as the co-generated hole quickly leaves the conjugated system[35]. We found spectroscopic evidences for a comparably stable electron-proton pair waiting for the following chemical processes to occur, while the photochemical one has been already completed. It is worth underlining that these spectroscopic changes are consistently recorded also at increasing $D_2O$ pressures (1

and 2 mbar pressure of $D_2O$) (Figs. S8, S9). As a reference experiment, we also exposed the bare gold substrate to $D_2O$ partial pressure and recorded only negligible shifts at the Au spectroscopic features, even upon light illumination, confirming that the shifts recorded for the carbon nitride are indeed related to the reaction cascade (Fig. S10).

## Evolution of $D_2$ and $O_2$

The process proceeds via the (thermochemical) formation of $D_2$ and $O_2$ as final products. These reactions involve the dimerization of

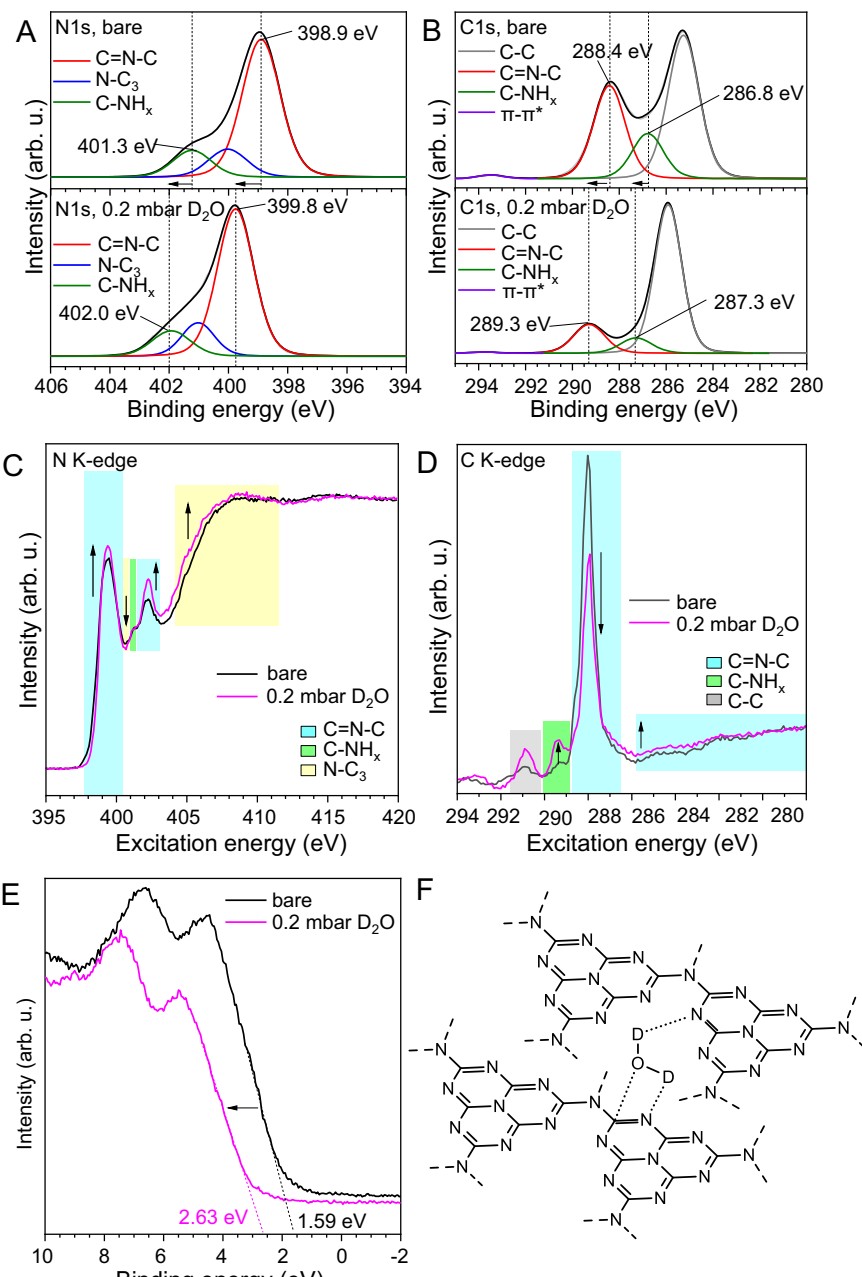

**Fig. 1 | In-situ adsorption of heavy water on the carbon nitride thin film surface. A** XPS, N1s spectra of carbon nitride thin film bare (upper panel) and after adsorption of heavy water (lower panel); arrows indicate spectral shift direction. **B** XPS, C1s spectra of carbon nitride thin film bare (upper panel) and after adsorption of heavy water (lower panel); arrows indicate spectral shift direction. **C** NEXAFS, N K-edge of the carbon nitride thin film bare (black line) and after adsorption of heavy water (magenta line); colored boxes highlight the different chemical features and the arrows indicate the spectral changes intensity. **D** NEXAFS, C K-edge of the carbon nitride thin film bare (black line) and after adsorption of heavy water (magenta line); colored boxes highlight the different chemical features and the indicate the spectral changes intensity. **E** VB-XPS of the carbon nitride thin film bare (black line) and after adsorption of heavy water (magenta line) with relative valence band values, arrow indicates spectral change direction. **F** Schematic representation of the carbon nitride-heavy water structure after adsorption.

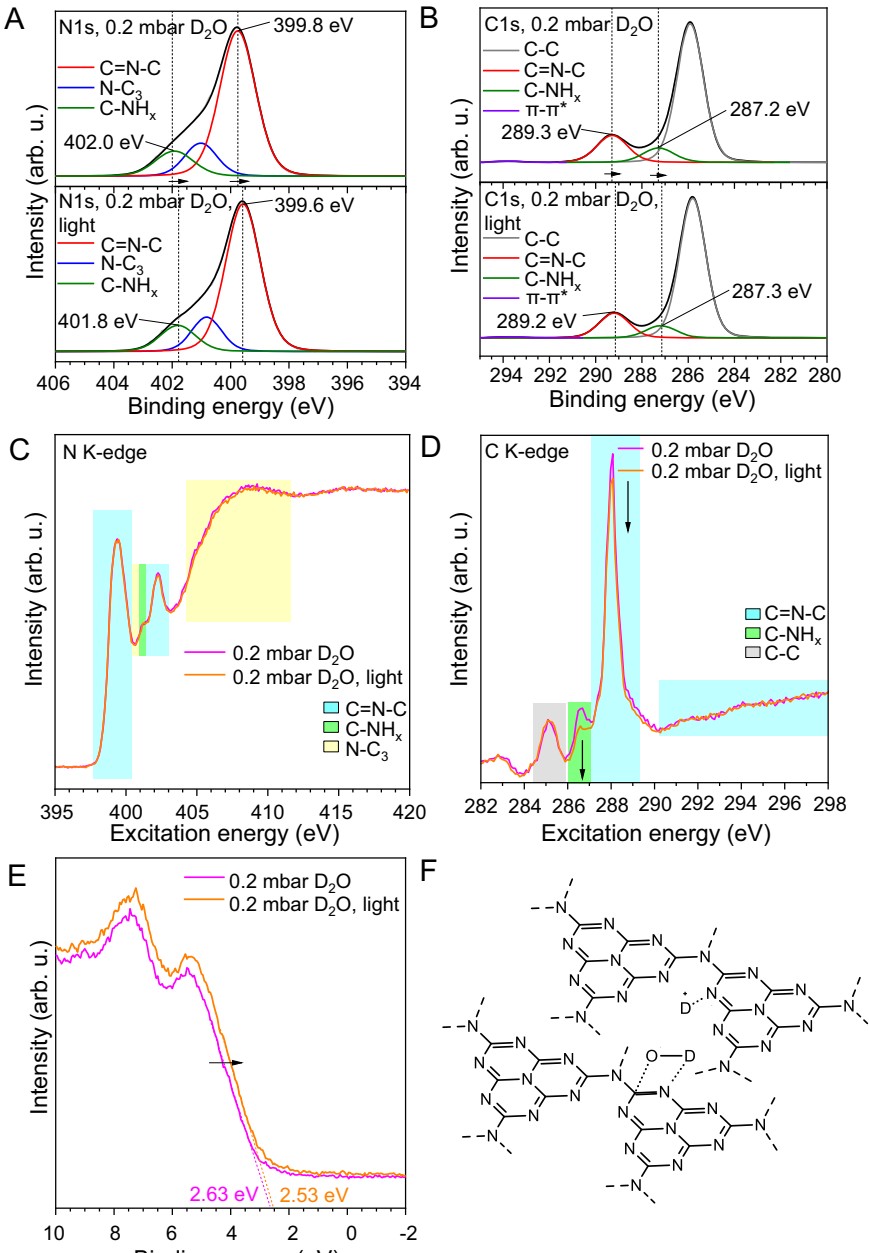

**Fig. 2 | In-situ illumination of the carbon nitride-heavy water system. A** XPS, N1s spectra of carbon nitride thin film after adsorption of heavy water (upper panel) and after illumination with the solar simulator (lower panel); arrows indicate spectral shift direction. **B** XPS, C1s spectra of carbon nitride thin film after adsorption of heavy water (upper panel) and after illumination with the solar simulator (lower panel); arrows indicate spectral changes direction. **C** NEXAFS, N K-edge of the carbon nitride thin film after adsorption of heavy water (magenta line) and after light illumination (orange line); colored boxes highlight the different chemical features. **D** NEXAFS, C K-edge of the carbon nitride thin film after adsorption of heavy water (magenta line) and after light illumination (orange line) with relative valence band values; colored boxes highlight the different chemical features; the arrow indicates intensity change. **E** VB-XPS of the carbon nitride thin film after adsorption of heavy water (magenta line) and after light illumination (orange line) with relative valence band values; arrow indicates spectral change direction. **F** Schematic representation of the carbon nitride-water structure after light illumination.

electron proton pairs or the corresponding disproportionation of $D_2O_2$ and are –without a metal cocatalysts- known to be comparably slow, in the thousand seconds range[11,23,36]. We thereby followed the evolution of $D_2$ and $O_2$ with a downstream TOF-MS detector, recording in subsequent cycles of light illumination on that timescale the presence of the products (Fig. 3). Upon illumination with the solar simulator (yellow boxes), the signal of the $D_2O$ precursor is significantly reduced, while a counterbalancing slow increase in the signals of the products $D_2$ and $O_2$ is observed. To confirm the thermochemical character of this step, the solar simulator was turned off, resulting in a

continued, but slowly decreasing formation of the products' TOF-MS signals while the signal of $D_2O$ increases sharply.

It is worth noticing that turning off the solar simulator illumination after the first cycle results in a consistent change also in the XPS and NEXAFS profiles resembling those recorded before first illumination step (Fig. S10). Further cycling makes the XPS and NEXAFS profiles change consistent to the first illumination cycle (Fig. S11). From the exponential increase of the products at the beginning of illumination and the exponential decline after stopping the illumination, it could be possible to determine the apparent time constants hydrogen

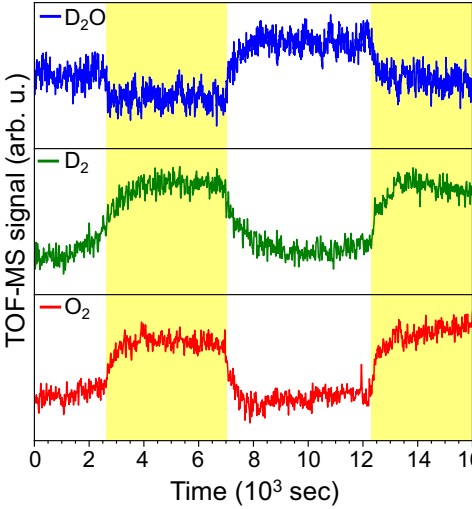

**Fig. 3 | In-situ photocatalytic evolution of $D_2$ and $O_2$.** TOF-MS signal of heavy water (upper panel), $D_2$ (middle panel), and $O_2$ (lower panel); white areas indicate the absence of illumination; yellow areas indicate the time the carbon nitride photocatalyst was subjected to illumination by means of the solar simulator.

dimerization and $D_2O_2$ decomposition, which is however beyond the scope of this work. We also estimated the conversion of $D_2O$ for two cycles to be about 30% and the relative yields with respect to $D_2$ (15% and 18% for cycle 1 and 2 respectively) and $O_2$ (13% and 14% for cycle 1 and 2 respectively). Despite the variability in these numbers, the results clearly point to a significant relative conversion of heavy water under solar simulator illumination. Indeed, when exposing the bare gold substrate to heavy water vapors and subsequently the illumination by means of the solar simulator, negligible variations in the signals of $D_2O$, $D_2$, and $O_2$ are recorded (Fig. S13).

## Discussion

In this study, we employed in-situ X-ray techniques to closely follow the mechanism of photocatalytic water splitting with a carbon nitride thin film. Step-by-step vapor exposure and light illumination enabled us to unravel the effects of heavy water on the carbon nitride electronic structure, providing experimental insights into one of the most extensively studied reactions of the last decade, both theoretically and experimentally.

Key outcomes from the experiments include the following observations for which we have recorded spectroscopic signatures during the experiments. First, the adsorption of water alters the surface electron density and electronic properties of the photocatalyst. Upon light absorption, we record a residual effective reduction of the overall electronic system, with a potential signature of a delocalized electron-proton couple integrated at the surface. However, we could not record the kinetically faster hole transfer from the carbon nitride to the water phase due to the different timescales of the photochemical oxidation process and X-Ray spectroscopies acquisition time. This fast cross-interface transfer of the photogenerated hole points to a potential carbon-oxygen back-donation mechanism. The transfer results in a surface bound hydroxyl radical, which is stabilized at a carbon site by the spin component of the localized photoelectron. The interaction of the carbon nitride with water and light produces specific spectroscopic signatures that point to different contributions for the different functional groups at each step. Finally, deuterium and oxygen are generated in a consecutive thermal chemical step with a conversion time in the thousand seconds timescale.

Moreover, the photosynthetic reaction was reproduced through two subsequent cycles of solar simulator illumination with consistent results. Since some of these observations were not considered in the

rich previous discussions of artificial photosynthesis as such, we consider them important for the further development of the field, potentially also well beyond the specific choice of the present catalyst-substrate system.

## Methods

### Chemicals

Melamine (99%) and heavy water (99.9 at% D), isopropanol (≥99.7%), methanol (≥99.8%), and toluene (99.8%) for the experiments were purchased from Sigma Aldrich and used without further purification. The gold substrates were purchased from Biolin Scientific and thoroughly rinsed with water, ethanol, and isopropanol prior to use.

### Synthesis of carbon nitride thin films

Carbon nitride thin films were prepared by chemical vapor deposition on a gold substrate (QSense, QSX 301). The preparation of carbon nitride thin films was conducted with a planarGROW-3S-OS CVD System for Organic Semiconductor, provided by planarTECH, with a 3 in. diameter quartz tube. The substrate is placed on a holder in the downstream oven and a glass boat containing melamine (5 g) in the upsteam one. The downstream oven temperature is raised to 550 °C at 10 Torr with 50 sccm nitrogen flow as carrier gas. Then, the upstream oven temperature is increased to 300 °C at a rate of 10 °C min⁻¹ and kept for 30 min. The substrates were kept for additional 120 min at 550 °C after the upstream oven program was finished. The samples were let cool down naturally and collected at room temperature.

### Characterization

Near ambient pressure (NAP)-XPS experiments were designed on the 1.5 GeV storage ring at MAX IV Laboratory in Lund, Sweden. We analyzed the samples and gold reference (Fig. S1) in the soft X-ray elliptically polarizing undulator and monochromated SPECIES beamline (30–1500 eV) for ambient pressure XPS branch A in a multi-bunch operation regime (Fig. S4)[37].

The C1s, N1s, and O1s core-level spectra were collected under ultra-high vacuum conditions (~$10^{-7}$ bar). Deuterated water ($D_2O$) and other solvents were degassed by several freeze-pump-thaw cycles before injection in the sample chamber. The sample chamber was equilibrated for 15 min to reach a steady state $D_2O$ signal prior spectroscopic acquisition. The spectroscopic characterizations were performed at increasing pressure of $D_2O$ (0.2, 1, and 2 mbar), where the pressure was monitored and adjusted during all the experiments, to avoid any significant fluctuation in the local concentration. Illumination cycles were performed using a HAL-320W solar simulator which mimics the solar spectrum Air Mass 1.5 G (350-1800 nm), and calibrated at the sample position inside the reaction cell (irradiance intensity 50 mW cm⁻², spot size 500 μm * 500 μm) for 30 min. We selected four different spectral lines (C1s, N1s, O1s, and Au4f) to characterize the chemical state at the sample surface (kinetic energy 150 eV) along with the VB-XPS region to evaluate the changes in the highest occupied molecular orbital (HOMO) electron density position in a quantitative manner. O1s XPS and NEXAFS spectra are reported in Fig. S5, for the bare carbon nitride thin film sample, Fig. S6 for the carbon nitride exposed to heavy water vapors, and Fig. S7 for the carbon nitride film exposed to heavy water and subjected to solar simulator illumination (cycle 1). The XPS overview of the sample exposed to different partial pressures of heavy water and in presence and absence of solar simulator illumination are available in Fig. S12. For the illumination cycles recording the evolution of the products using the carbon nitride thin film (Fig. 3) the light was turned on after 2640 s, turned off at 7020 s, and turned on again at 12240 s. We additionally test the evolution of the products for the bare gold substrate (Fig. S13) in presence and absence of solar simulator illumination.

The effect of different solvents has been tested by dosing isopropanol, toluene or methanol into reaction chamber under a

0.2 mbar pressure. The sample chamber was equilibrated for 15 min to reach a steady state solvent signal prior spectroscopic acquisition. Spectral lines (C1s, N1s, O1s) and VB-XPS region were collected using kinetic energy 150 eV. Desorption of solvents was performed by evacuating the reaction chamber up to pressure $1 \cdot 10^{-8}$ bar for 40 min prior to collecting the data.

XPS data were analyzed using CASA XPS software using a Shirley-type background subtraction, unless otherwise stated. Detailed summary of the deconvolution parameters for all the samples can be found in Supplementary Tables. Peak deconvolution and attribution were done in agreement with previously reported data on carbon nitride materials[18,20,38–40]. HOMO values were obtained as interpolation of the linear area of the VB-XPS spectra edges. NEXAFS spectra were normalized after the background subtraction using ATHENA package and elaborated according to previous reports on carbon nitride materials[20,21].

To investigate radical formation, Electron Paramagnetic Resonance (EPR) measurements were performed on a reaction mixture containing the DMPO as a spin trap. The spin trap was added to 2 mL of $D_2O$ (99.9% D) in a glass vial coated with a thin film of carbon nitride. The reaction mixture was then illuminated by a purple LED lamp under constant stirring. After 40 min intervals, a 50 µL aliquot was collected, sealed in glass capillaries, and analyzed by EPR inside a quartz tube (Wilmad Labglass). For the experiments, 10 mg of DMPO was used. The adducts were identified by simulating the EPR spectra using Easyspin® software.

Atomic force microscopy analysis has been conducted on NanoScope-Multimode AFM (Digital Instruments) atomic force microscope.

## Data availability
Source Data file has been deposited in EDMOND under accession code[41].

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

## Acknowledgements

This work was supported by European Union's Horizon research and innovation program (CATART, grant agreement number 101046836, beneficiaries O.S. and P.G.). The materials presented and views expressed here are the responsibility of the authors(s) only. The EU Commission takes no responsibility for any use made of the information set out. We acknowledge MAX IV Laboratory for time on Beamline Species under Proposals 20230810, 20230220, 20221567, 20241034. Research conducted at MAX IV, a Swedish national user facility, is supported by the Swedish Research council under contract 2018-07152, the Swedish Governmental Agency for Innovation Systems under contract 2018-04969, and Formas under contract 2019-02496. The authors thank Margit Andersson, Esko Kokkonen, and Alexander Klyushin for the technical support during the development of the project. We would like to thank Dr. Xue Zhang for help in performing AFM analysis and Prof. Iver Lauermann for access to UPS. The authors thank the Max Planck Society for financial support. D.C. acknowledges the Fritz Haber Institute (FHI) of the Max Planck Society for institutional support and the German Federal Ministry for Education and Research (BMBF, grant Catlab 03EW0015B) for funding.

## Author contributions

D.C. and S.Z. contributed equally to this work. P.G. and D.C. conceived the idea of this work. D.C., O.S., P.G discussed and planned the experiments. D.C. and S.Z. conducted the experiments and analyzed the data. P.G. supervised the project. S.Z., M.A. and P.G. wrote the manuscript. All authors discussed the results and contributed to the final manuscript.

## Funding

## Competing interests

The authors declare no competing interests.
