## [Peer Review File · Nature Communications]

Carbon nitride caught in the act: unexpected roles of surface interactions in artificial photosynthesis

Corresponding Author: Dr Paolo Giusto

Version 0:

Reviewer comments:

Reviewer #1

(Remarks to the Author)

This manuscript presents a detailed study on the role of surface interactions in photocatalytic processes using synchrotron-based in situ spectroscopies. The authors investigate electron density changes at the surface of carbon nitride upon heavy water adsorption w/ and w/o light, which mimics photocatalytic reactions. The manuscript includes comprehensive experimental data, such as in situ XPS, NEXAFS, and TOF-MS spectra, which are well-documented and provide clear evidence for the proposed mechanisms.

I would recommend the publication of this paper, provided the authors address the following items:

1. Mechanistic discussion: The discovery that water adsorption significantly alters the electronic structure of carbon nitride is very interesting. However, the manuscript lacks a detailed mechanistic discussion on how these surface interactions facilitate the photocatalytic process. Incorporating computational modeling could strengthen this section. Additionally, it would be beneficial to explain how these findings can inform the design of more efficient catalysts for water splitting in photocatalysis.
2. Effect of D₂O pressure: The authors varied the pressure of heavy water. However, the manuscript does not sufficiently discuss the effect of D₂O pressure on the electronic properties of carbon nitride. Detailed results on this aspect would enhance the understanding of the pressure-dependent behavior. It would also be useful to know if the spectra were recorded after the complete removal of vapor-phase D₂O. Would the observed changes in electronic structure remain post desorption? Monitoring the redox properties of carbon nitride upon water adsorption and desorption would provide valuable insights.

Reviewer #2

(Remarks to the Author)

In this work, the authors studied how the interaction between carbon nitride and D₂O affects the electronic state of carbon nitride. The study is interesting, but more needs to be done to confirm the authors' conclusions.

1. Maybe I misunderstood the authors' meaning: "...it will be shown here that, contrary to our simplistic initial expectations, the presence of water already alters the electronic structure of the photoactive carbon nitride semiconductor...".

It is well known that solvents change the electronic structure of semiconductors. Solvents can modify the surface states, interact with the surface atoms of semiconductors, and alter their electronic states. This can lead to changes in the surface band structure, potentially creating or removing surface trap states.

The dielectric constant of the solvent affects the screening of charge carriers in the semiconductor. Some solvents may donate or accept electrons from the semiconductor, leading to doping-like effects. This can shift the Fermi level and alter the carrier concentration in the material.

2. Can the authors change the solvent and see how it affects the properties of the carbon nitride?
3. The authors should give more details about the carbon nitride film. What is the thickness? Is it porous or dense? Does the water can penetrate only on the interface? If it is a thin layer – how does the Au change the fermi level?
4. Is it unclear what they mean by writing that they have a joint new hybrid semiconductor structure? If the effect is only on the surface, how does it change the bulk?
5. Why does the ratio between the C-C and C-N=C change with D₂O? I can understand the shift, not the population change.
6. I would appreciate some statistics in the data as the changes are so small. 0.1 eV (Figure 2A-B) is within the error bar of the instrument.
7. The formation of D₂ and O₂ is not clear. Is it due to full water splitting? Carbon nitride from melamine is considered non-

active for full water splitting without a catalyst.

8. There are too many speculations and unsupported claims in the manuscript.

Reviewer #3

(Remarks to the Author)

The authors present work on carbon nitride surface interactions in artificial photosynthesis using advanced synchrotron facility. The work presents a study on the photocatalytic activation of a thin film of polymeric carbon nitride using heavy water (D_2O). The authors employed advanced spectroscopic techniques (XPS, NEXAFS) to investigate the electronic modifications induced by the adsorption of D_2O on the material. The results indicate a strong interaction between water and carbon nitride, leading to a significant change in the material's electronic structure. This change translates into an increase in the catalyst's oxidation potential, thus facilitating the initial step of artificial photosynthesis: water oxidation. The manuscript is well written and organized and results are clearly discussed. The topic is very interesting as the artificial photosynthesis is of high importance. The novelty of the findings is based on the substantial shift, not observed before, in the electronic structure of carbon nitride upon water adsorption and on the proposal of a new mechanism involving proton-coupled electron transfer and the formation of a stable intermediate. The work could be published in Nature Comm after some minor revision. My remarks are:

1. Deconvolution parameters should be given. What kind of constraints were used? Why the FWHM changes only for N-C3 after adsorption of water? It is also observed for C1s components.
2. How can the shift of XPS peaks to higher binding energies after D_2O adsorption be explained? Is it solely due to a charging effect or to a real modification of the atoms' chemical environment?
3. Authors stated that the hole created by the photochemical excitation is transferred to the heavy water, forming a surface $DO\cdot$ radical. But there is no proofs for that. Figure 2F shows only a schematic representation. The formation of these radicals could be observed in other experimental measurements such as EPR/lights.
4. Concerning the mechanism. What is the exact mechanism of formation of the N-photoelectron $\cdots D^+$ and N $\cdots D-O\cdot\cdots C$ species? How can their existence be confirmed by other experimental techniques?
5. How does the catalyst's stability evolve over time under light irradiation? Are there any signs of deactivation or modification of the material's structure?
6. What other semiconductor materials could exhibit similar behaviors? To confirm these findings.

Version 1:

Reviewer comments:

Reviewer #1

(Remarks to the Author)

I believe the authors have adequately addressed my comments, and I recommend this manuscript for acceptance.

Reviewer #2

(Remarks to the Author)

The authors addressed the remarks.

Reviewer #3

(Remarks to the Author)

The authors have replied to all referee questions and remarks. They have improved their manuscript. I recommend now the article for publication.

Reviewers' comments in black – Authors' responses in blue

Reviewer #1

Q1: Mechanistic discussion: The discovery that water adsorption significantly alters the electronic structure of carbon nitride is very interesting. However, the manuscript lacks a detailed mechanistic discussion on how these surface interactions facilitate the photocatalytic process. Incorporating computational modeling could strengthen this section. Additionally, it would be beneficial to explain how these findings can inform the design of more efficient catalysts for water splitting in photocatalysis.

The authors thank this Reviewer for their insightful suggestion regarding the inclusion of a more detailed mechanistic discussion and the potential role of computational modeling in our study. We appreciate the significance of these aspects in deepening the understanding of how water adsorption impacts the electronic structure of carbon nitride and its implications in photocatalysis. Our current study focuses on the experimental observations of water adsorption on the carbon nitride surface and how this impacts the photocatalyst electronic properties facilitating the otherwise difficult step of photocatalytic water splitting. The current in-operando photocatalytic experiments reveal that the adsorption of water causes a depletion of the electron density at the photocatalyst surface, conveyed by a significant shift of the valence band value, as highlighted by the VB-XPS data, enabling the water splitting process. Theoretical studies on the carbon nitride photocatalytic water splitting have previously shown results that are consistent with our experimental ones, e.g. Phys. Chem. Chem. Phys., 2014, 16, 15917-15926, J. Phys. Chem. Lett. 2018, 9, 16, 4695–4699, Phys. Chem. Chem. Phys., 2014, 16, 3299-3304, J. Phys. Chem. A 2017, 121, 25, 4754–4764. For example, Wu et al. (Phys. Chem. Chem. Phys., 2014, 16, 3299-3304) have studied how the water interacts with the carbon nitride at increasing coverage and have shown how the adsorption causes a shift of the valence band energy towards higher values (+2.50 eV). While we recognize the value of incorporating additional computational models for depicting the detailed underlying mechanisms of these surface interactions and how they facilitate the photocatalytic process, we believe that this would go beyond the scope of this manuscript. We expect that the current findings lay the necessary groundwork for future additional theoretical explorations, which will be part of our forthcoming work including, for instance, descriptors to achieve superior photocatalytic performances.

Q2: Effect of D₂O pressure: The authors varied the pressure of heavy water. However, the manuscript does not sufficiently discuss the effect of D₂O pressure on the electronic properties of carbon nitride. Detailed results on this aspect would enhance the understanding of the pressure-dependent behavior. It would also be useful to know if the spectra were recorded after the complete removal of vapor-phase D₂O. Would the observed changes in electronic structure remain post desorption? Monitoring the redox properties of carbon nitride upon water adsorption and desorption would provide valuable insights.

We would like to thank the reviewer for the insightful comment that helps us improving the quality of our work and the understanding of the role of vapor pressure on the electronic properties of carbon nitride thin films. At first, the XPS and VB-XPS of the carbon nitride thin films have been plotted at increasing pressure, to emphasize the effect of the water adsorption on the electronic properties (Figure R1 a-c).

In the first set of results reported in the main manuscript and SI, we recorded the XPS, VB-XPS, and XANES spectra of the CN thin film exposed to increasing D₂O pressure values (0.2-2 mbar). It was previously shown by Wu et al. (Phys. Chem. Chem. Phys., 2014, 16, 3299-3304), that the adsorption of water at the carbon nitride occurs via the formation of hydrogen bonds and, at low coverage, is thermodynamically favored with respect to the formation water clusters. At lower coverage, occurring at lower D₂O vapor pressures, the water preferentially adsorbs at the carbon nitride surface with the heptazine's N acting as hydrogen bond donor and the negatively polarized O of the heavy water back-binds with the positively polarized heptazine's C (Phys. Chem. Chem. Phys., 2014,16, 15917-15926). At higher coverage, the water forms clusters on the surface of the carbon nitride, stabilized by hydrogen bonds between the water molecules (Phys. Chem. Chem. Phys., 2014, 16, 3299-3304). To investigate the role of the adsorption of heavy water molecules on the carbon nitride electronic properties we merged the XPS and VB-XPS data stacked as a function of pressure to enable a more direct comparison between the different conditions. As shown in Figure R1a and b, a major shift of the C1s and N1s features towards higher binding energies is recorded from the bare sample the lowest heavy water vapor pressure (0.2 mbar). Notably, the elemental peaks and the deconvoluted features do not significantly shift at higher heavy water pressures, except for the C1s C-N=C feature. Here, the peak maxima at higher vapor pressures (1 and 2 mbar, 2nd and 3rd panels, respectively, Figure R1a) are shifted to lower binding energies of about 0.3 eV with respect to the 0.2 mbar D₂O pressure. We attribute this effect to the formation of a multilayer/cluster adsorption of water occurring when all the adsorption sites on the carbon nitride are occupied. As a result, the donation of electron density from the heavy water's O to the C of the heptazine units is reduced causing a shift to lower binding energies of the C-N=C feature in the C1s, pointing to stronger interactions for the heavy water intermolecular (O---D) than the one between heavy water and carbon nitride (C---O). While the multilayer adsorption affects the C1s spectrum at 1 and 2 mbar pressure with respect to the 0.2 mbar one, this has only a minor effect on the N1s, which is attributed to the stronger interactions in the hydrogen bond formation between the heptazine's N and the heavy water's D (N---D) with respect to heavy water intermolecular ones (O---D). This is further supported by the VB-XPS results, where at 0.2 heavy water vapor pressure the resulting VB value is shifted from 1.59 eV to 2.63 eV. The increase of vapor pressure causes an increase of this value up to 2.80 eV. This is consistent with our interpretation, since the formation of water clusters (or multilayer adsorption) reduces the back-binding effect between the heavy water's O and the heptazine's C, further reducing the electron density at the carbon nitride surface.

The effect of D₂O pressure on the electronic properties of carbon nitride thin film is now discussed in detail in Supplementary note 1. Moreover, in the following we expanded the study by additional

experiments on the electronic properties after the desorption of water from the surface of the carbon nitride thin film.

Figure R1. N1s (A), C1s (B), VB-XPS (C) spectra of carbon nitride thin films at increasing D_2O vapor pressure. From top to bottom: bare carbon nitride film (top panel), 0.2 mbar D_2O (second panel), 1 mbar D_2O (third panel), 2 mbar D_2O (fourth panel).

The authors applied and were granted further beamtime to perform an additional set of experiments and study the effect of the water desorption at the lowest vapor pressure (0.2 mbar heavy water). In the following we report the results of our experiment on the electronic properties and characteristics XPS features shifts of the carbon nitride film (Figure R2). As depicted in Figure 2 (main manuscript) upon water adsorption, the formation of hydrogen bonds at the carbon nitride surface is primarily driven by interactions between nitrogen atoms (acting as electron donors) and water molecules. As a result, the electron density around nitrogen is reduced, causing the N1s and C1s features to shift towards higher binding energies also in this set of experiments and in good agreement with our previous results. The water desorption step has been performed by stopping the heavy water flow and evacuate the measurement chamber in vacuum for 45 minutes before collecting the XPS spectra. After the chamber evacuation and water desorption, we record a negligible shift in the N1s spectra (<0.1 eV). However, in the C1s spectra we record a significant shift towards higher binding energies of <0.3 eV. This is attributed to the removal of more loosely bound water from the surface of the carbon nitride where only the strongest bound water molecules are retained, further depleting the electron density at the carbon. The resulting VB is also reduced and attributed to the formation of the most stable heavy water-carbon nitride system, with a VB

value recorded of 2.8 eV. The results are consistent with the investigations of Wu et al. (*Phys. Chem. Chem. Phys.*, 16, 3299-3304 (2014)), revealing changes in the electronic properties of water, with the experimental results showing the same trend and consistent change in the electronic properties. Indeed, in their work, Wu et al. found that the adsorption of water induces a structural (and irreversible upon desorption) change in the carbon nitride from a planar to a buckled structure, resulting in a material with a higher oxidation potential. In their calculation, the adsorption of water leads to a higher VB values, up to 2.5 eV for a single water molecule adsorbed per unit cell.

Figure R2. XPS spectra of N1s (A) and C1s (B) and VB-XPS (C) of carbon nitride under flow of 0.2 mbar D₂O (middle panel) and after desorption (bottom panel).

Furthermore, XPS at increasing kinetic energy (150 keV, 400 keV and 800 keV, corresponding to 0.51 nm, 0.93 nm, and 1.5 nm penetration) spectra were collected to evaluate the effect of the water adsorption at different penetration depths on the carbon nitride thin film exposed to 0.2 mbar of heavy water. In Figure R3 A, we record a shift towards lower binding energies of about 0.3 eV. This points to a stronger effect of the water adsorption on the electronic properties of the carbon nitride thin film surface rather than on the subsurface. This is further confirmed by the shift recorded at the C1s features (Figure R3 B) which shift towards lower binding energies of about 0.2 eV with respect to the lowest kinetic energy applied. Furthermore, at increasing kinetic energies, we can clearly see an increase in the areal ratio between the C-N=C features and the C-C features (attributed to the adventitious carbon) confirming that the C-C component is to be attributed to surface contamination of carbon species adsorbed on the surface. Eventually, we confirmed that the electronic properties, in terms of valence band, are significantly affected on the surface by adsorption of water and the effect is less pronounced at higher depths (Figure R3 C). Indeed, upon water adsorption the carbon nitride thin film surface valence band shifts towards higher values (2.6 eV) while at lower penetration depths it retains consistent values with the bare sample (1.6 eV) and supporting the dominant effect of water at the surface level.

Figure R3. XPS spectra of N1s (A), C1s (B), and VB-XPS (C) measured at kinetic energy 150 eV (top panel) 400 eV (middle panel) and 800 eV (bottom panel) under flow of 0.2 mbar D₂O.

Reviewer #2

Q1: Maybe I misunderstood the authors' meaning: "...it will be shown here that, contrary to our simplistic initial expectations, the presence of water already alters the electronic structure of the photoactive carbon nitride semiconductor...". It is well known that solvents change the electronic structure of semiconductors. Solvents can modify the surface states, interact with the surface atoms of semiconductors, and alter their electronic states. This can lead to changes in the surface band structure, potentially creating or removing surface trap states.

The dielectric constant of the solvent affects the screening of charge carriers in the semiconductor. Some solvents may donate or accept electrons from the semiconductor, leading to doping-like effects. This can shift the Fermi level and alter the carrier concentration in the material.

We would like to thank the reviewer for pointing this out. Indeed, we agree with the reviewer that the statement in the current phrasing may lead to misunderstandings. We acknowledge that the adsorption of small molecules on the surface of semiconductors leads to change of the surface electronic properties (e.g., *The Journal of Physical Chemistry C* 121 (18), 9815-9824 (2017), *J. Phys. D: Appl. Phys.* 51 433001 (2018), *Nature Materials* 2, 253–258 (2003), *Advanced Functional Materials*, 2406528 (2024)). However, here we wanted to refer to the case of the water splitting with carbon nitride, where the effect of water adsorption on the surface electronic properties has been explored, to the best of our knowledge, only at a theoretical level (e.g. *Phys. Chem. Chem. Phys.*, 16, 15917-15926 (2014), *J. Phys. Chem. Lett.*, 9, 16, 4695–4699 (2018), *Phys. Chem. Chem. Phys.*, 16, 3299-3304 (2014), *J. Phys. Chem. A* 121, 25, 4754–4764 (2017)). With this sentence, we aimed to emphasize that the change in the electronic properties recorded play a pivotal role in the photocatalytic process. In order to avoid any misunderstandings, we rephrased the sentence in the manuscript as follows: "For instance, it will be shown that the water adsorption

step plays a pivotal role by altering the surface electronic properties of the photoactive carbon nitride semiconductor, creating an activated state through pre-polarization”

Following on the second part of this reviewer comment and the following question and we applied and were granted additional beamtime to further investigate this phenomenon by examining the effects of various solvents on the electronic structure of carbon nitride by means of AP-XPS. These findings are detailed in Supplementary Note 2 and presented below as a response to Q2.

Q2: Can the authors change the solvent and see how it affects the properties of the carbon nitride?

We would like to thank the reviewer for this insightful remark. We acknowledge that such an investigation is essential for a comprehensive understanding of how solvents impact the material's electronic properties. For this reason, we have studied the effect of three additional solvents with different dielectric constant (ϵ) (methanol, isopropanol, and toluene) on the electronic properties of the carbon nitride thin films on the electronic properties of the carbon nitride thin film. However, due to the relevance of this study we plan to expand this work in our upcoming studies, as we believe it will yield valuable insights into the semiconductor behavior of carbon nitride in different solvent environments and reactions.

We studied the effect of the following solvents with different ϵ (methanol ($\epsilon=32.7$), isopropanol ($\epsilon=19.9$), and toluene ($\epsilon=2.4$), in addition to D_2O ($\epsilon=78.1$ at $25^\circ C$) (The Journal of Physical Chemistry 71(3), 656-662 (1967)) at the pressure of 0.2 mbar on the electronic properties of carbon nitride. The recorded XPS spectra are shown in Figs. R4-R6.

Upon methanol adsorption (Figure R4), we observe a shift in the N1s features towards higher binding energies (0.5 eV for C-N=C, 0.4 eV for N-C₃, and 0.2 eV for C-NH_x), indicating electron density donation from nitrogen atoms to the methanol. This occurs due to hydrogen bond formation between the hydrogen atom of the hydroxy group (acceptor) in methanol and the nitrogen atoms (donors) of the carbon nitride. Consequently, the overall electron density of the semiconductor decreases, as reflected by the shift in the carbon spectroscopic features towards higher binding energies. Additionally, the arising of an O-H feature in the O1s spectrum is observed. The electron density donation into hydrogen bonds with methanol also causes a shift in the valence band position of the carbon nitride film by approximately 0.3 eV, lower than that of heavy water. This indicates that polar solvents like methanol can form a hybrid semiconductor structure with carbon nitride, resulting in significant electron density redistribution. We attribute the lower shifts recorded here to the lower dielectric constant of the methanol with respect to (heavy) water, inducing a weaker displacement of the electron density of the carbon nitride surface.

Figure R4. XPS spectra of N1s (A) C1s (B) O1s (C) and VB-XPS (D) of bare carbon nitride (top panel) and carbon nitride under flow of 0.2 mbar methanol (middle panel).

For the isopropanol adsorption study, lower shifts were observed in the C1s and N1s spectra (Figure R5 A, B). In the N1s spectrum (Figure R5 A), the N-C3 feature shifted approximately 0.2 eV towards higher binding energy upon isopropanol adsorption, due to electron density donation from the carbon nitride's nitrogens that act hydrogen bond donor with the H(O)- of the isopropanol. The adsorption of isopropanol is also evident from a decrease in the relative intensity of C-N=C features with respect to the C-C one, in the C1s spectrum, due to the C-C bonds present in the adsorbed molecules. In the O1s spectrum, we record the arising of an additional feature from the introduction of additional oxygen-containing species, e.g. O-H groups, on the carbon nitride surface. Moreover, changes in the VB-XPS spectrum reveal a shift of approximately 0.3 eV towards higher binding energy after adsorption, attributed to the formation of hydrogen bonds and the consequent donation of electron density from the carbon nitride to the isopropanol. Notably the shift is similar to that of methanol potentially attributed to the similar functionalities and relatively close ϵ values as compared to heavy water. These observations support the statements that the interaction of carbon nitride with solvents, especially those that can act as hydrogen bond donor, is not limited to water and it influences its surface electronic properties. Furthermore, the XPS shifts induced by the solvent are in good agreement with the solvent dielectric constant,

however, further investigations and a larger scope of solvents will be necessary to confirm this for our system.

Figure R5. XPS spectra of N1s (A) C1s (B) O1s (C) and VB-XPS (D) of bare carbon nitride (top panel) and carbon nitride under flow of 0.2 mbar isopropanol (bottom panel).

The introduction of toluene vapors into the system (Figure R6 A-C) causes noticeable changes in the N1s spectra, where the carbon nitride features shifted by approximately 0.2 eV towards higher binding energies. This is likely due to weak van der Waals interactions and π - π stacking between the π -electron system of toluene and nitrogen atoms in the carbon nitride matrix. These interactions reduce the electron density around the nitrogen sites, leading to the observed binding energy shifts. A similar shift is observed in the VB-XPS spectrum of about 0.2 eV. The shifts in binding energies can also be attributed to the polarizability of toluene's π -electron cloud, causing electron density redistribution in the triazine ring. However, we do not record significant shifts in the C1s spectrum, besides a change in the C-C to C-N=C population as a result of the toluene surface adsorption suggesting that the electron density at carbon atoms in the carbon nitride structure are less affected by the toluene for which the adsorption doesn't occur as a result of the formation of hydrogen bonds, but from weaker interactions such as π - π interactions.

Figure R6. XPS spectra of N1s (A) C1s (B) and VB-XPS (C) of bare carbon nitride (top panel) and carbon nitride under flow of 0.2 mbar toluene (bottom panel).

Eventually, solvents with higher dielectric constant and -OH groups can form hydrogen bonds with carbon nitride, leading to more pronounced changes on the surface electronic properties of the carbon nitride. On the other hand, in the case of toluene, a low dielectric constant solvent, a smaller change of the surface electronic properties has been also recorded, however, attributed to a different mechanism in the surface adsorption and attributed to weaker interactions, such as π - π interactions with the carbon nitride surface.

Q3. The authors should give more details about the carbon nitride film. What is the thickness? Is it porous or dense? Does the water can penetrate only on the interface? If it is a thin layer – how does the Au change the fermi level?

We thank the reviewer for raising this point. The carbon nitride film has thickness around 60 nm, as shown by AFM measurements (Figure R7). We confirm the film to be dense at a macroscale level, as previously shown in several reports by our group (e.g. *Chemistry of Materials*, 32(17), 7284-7291 (2020); *Advanced Materials*, 32, 1908140 (2020); *ACS Catalysis* 11(17), 11109-11116 (2021)) but containing structural pores in the heptazine structure.

Figure R7. AFM step-height measurement depicting the thickness of the prepared carbon nitride film.

For this reason, we expect that the water molecules can be stabilized within the pores between the heptazine units also in the subsurface. Indeed, we record a change in the electronic properties of the carbon nitride also at higher penetration depths, although lower than the one occurring on the upper surface (Figure R3). We copy in the following the relevant part from Reviewer#1 Q2:

“Furthermore, XPS at increasing kinetic energy (150 keV, 400 keV and 800 keV, corresponding to 0.51 nm, 0.93 nm, and 1.5 nm penetration) spectra were collected to evaluate the effect of the water adsorption at different penetration depths on the carbon nitride thin film. In Figure R3 A, we record a shift towards lower binding energies of about 0.3 eV. This points to a stronger effect of the water adsorption on the electronic properties of the carbon nitride thin film surface rather than on the subsurface. This is further confirmed by the shift recorded at the C1s features (Figure R3 B) which shift towards lower binding energies of about 0.2 eV with respect to the lowest kinetic energy applied. Furthermore, at increasing kinetic energies, we can clearly see an increase in the areal ratio between the C-N=C features and the C-C features (attributed to the adventitious carbon) confirming that the C-C component is to be attributed to surface contamination of carbon species adsorbed on the surface. Eventually, we confirmed that the electronic properties, in terms of valence band, are significantly affected on the surface by adsorption of water and the effect is less pronounced at higher depths (Fig.R3 B). Indeed, upon water adsorption the carbon nitride thin film surface valence band shifts towards higher values (2.6 eV) while at lower penetration depths it retains consistent values with the bare sample (1.6 eV) and supporting the dominant effect of water at the surface level.

Figure R3. XPS spectra of N1s (A), C1s (B), and VB-XPS (C) measured at kinetic energy 150 eV (top panel) 400 eV (middle panel) and 800 eV (bottom panel) under flow of 0.2 mbar D₂O.”

Additionally, we performed ultraviolet photoelectron spectroscopy (UPS) measurements on carbon nitride films deposited on a gold substrate, as used in our study, as well as on carbon and silicon substrates (see figure below) and evaluated the Fermi level. The presented spectra were normalized relative to the zero values near the valence band and calibrated in reference to an Ag standard. In the following discussion, we will evaluate the work function values, determined from the secondary electron cutoff (SECO) in the UPS spectra, which represent the energy difference between the Fermi level and the vacuum level (*Applied Surface Science Advances*, 13, 100384, (2023)). The work function of carbon nitride deposited on gold is in good agreement with the value previously reported by our group (*Advanced Materials*, 32, 1908140 (2020)). It is important to note that a charging effect was observed in the spectrum of the film deposited on the carbon substrate; hence, the initial slope was considered in the work function calculations. The lower measured work function on silicon (4.1 eV) is attributed to potentially interaction between the semiconductors leading to a slightly lower electron density on the surface of the carbon nitride film.

Figure R8. SECO region of UPS spectrum measured for carbon nitride thin film deposited on gold, carbon and silicon substrates.

Q4: Is it unclear what they mean by writing that they have a joint new hybrid semiconductor structure? If the effect is only on the surface, how does it change the bulk?

We appreciate the reviewer’s valuable remark. With the phrasing “joint new hybrid semiconductor”, we refer to the heavy water-adsorbed carbon nitride which has significantly different electronic properties with respect to the bare carbon nitride film prior to the exposure to the heavy water vapors. To explore if the effect of the water is limited to the surface or also affects the subsurface, we have expanded the dataset and added in supplementary information a detailed discussion on the effects of the heavy water on both the surface and subsurface electronic structure of carbon nitride. To investigate these effects, we performed AP-XPS with depth profiling at increasing kinetic energies, allowing us to analyze the properties at increasing depths of the carbon nitride film. Therefore, would like to recall the relevant section on the previous question of this reviewer (Q3) for the discussion on the electronic properties and XPS analysis at different penetration depths:

“Furthermore, XPS at increasing kinetic energy (150 keV, 400 keV and 800 keV, corresponding to 0.51 nm, 0.93 nm, and 1.5 nm penetration) spectra were collected to evaluate the effect of the water adsorption at different penetration depths on the carbon nitride thin film. In Figure R3 A, we record a shift towards lower binding energies of about 0.3 eV. This points to a stronger effect of the water adsorption on the electronic properties of the carbon nitride thin film surface rather than on the subsurface. This is further confirmed by the shift recorded at the C1s features (Figure R3 B) which shift towards lower binding energies of about 0.2 eV with respect to the lowest kinetic energy applied. Furthermore, at increasing kinetic energies, we can clearly see an increase in the

areal ratio between the C-N=C features and the C-C features (attributed to the adventitious carbon) confirming that the C-C component is to be attributed to surface contamination of carbon species adsorbed on the surface. Eventually, we confirmed that the electronic properties, in terms of valence band, are significantly affected on the surface by adsorption of water and the effect is less pronounced at higher depths (Fig.R3 B). Indeed, upon water adsorption the carbon nitride thin film surface valence band shifts towards higher values (2.6 eV) while at lower penetration depths it retains consistent values with the bare sample (1.6 eV) and supporting the dominant effect of water at the surface level.

Figure R3. XPS spectra of N1s (A), C1s (B), and VB-XPS (C) measured at kinetic energy 150 eV (top panel) 400 eV (middle panel) and 800 eV (bottom panel) under flow of 0.2 mbar D₂O.”

Q5: Why does the ratio between the C-C and C-N=C change with D₂O? I can understand the shift, not the population change.

The authors thank the reviewer for raising this valuable point. In the elaboration of the data, we noticed an increase in the signal of the C-C bonds which are attributed to the adventitious carbon contribution, as also demonstrated by the depth profiling analyses. Here, upon the adsorption of heavy water, the water is expected to interact predominantly with the carbon nitride therefore reducing its relative intensity, while not affecting the one of the surface contaminations (adventitious carbon), resulting in an apparent increase of the C-C signal population.

Q6: I would appreciate some statistics in the data as the changes are so small. 0.1 eV (Figure 2A-B) is within the error bar of the instrument.

We thank the reviewer for this important point on the statistical significance of the shifts we recorded. It is worth mentioning that the measurements reported with indicated shifts have all been done during the same run, without removing the sample from the measurement chamber. The nominal spectrometer resolution is 0.1 eV and, therefore values in the figures in the manuscript as well as in supplementary have been corrected accordingly. However, to provide more background, we checked the changes in the binding energy values and intensities of the C1s and N1s peaks. Here, we present 20 dark/light spectra for both N1s and C1s. To assess the statistical significance,

we tracking the C=N-C sp^2 component in both C1s (~289eV) and N1s(~399). For C1s the mean shift is 0.15eV with a standard deviation of 0.016eV, while for N1s, the mean shift is around 0.12eV with a standard deviation 0.027eV. These findings suggest that the observed shift has are statistically significant.

Figure. R9. Statistical analysis of data based on C-N=C sp^2 component of N1s and C1s core spectra.

Q7: The formation of D₂ and O₂ is not clear. Is it due to full water splitting? Carbon nitride from melamine is considered non-active for full water splitting without a catalyst.

We appreciate the reviewer's comment and we agree that previous reports have shown that the carbon nitride from melamine is not active for the direct evolution of D₂ and O₂ from water in absence of an additional catalyst. In our study, we clearly demonstrated that, under specific conditions carbon nitride can indeed catalyze the production of D₂ and O₂ via a proton-coupled electron transfer (PCET) mechanism. The films prepared by chemical vapor deposition are highly

homogeneous and, supported by the absence of any signal of gold from spectroscopic measurements, we can confidently assume that the substrate does not play any active role in the photocatalytic process. However, the gold substrate was necessary to avoid any excessive surface charge during the experiment and ensure the reproducibility of the results reported. Our findings indicate that the mechanism involves the oxidation of heavy water with the formation of D₂O₂ as an intermediate product which subsequently decomposes, resulting in the generation of D₂ and O₂, which has been postulated in several theoretical studies, e.g. Phys. Chem. Chem. Phys., 2014, 16, 15917-15926, J. Phys. Chem. Lett. 2018, 9, 16, 4695–4699, Phys. Chem. Chem. Phys., 2014, 16, 3299-3304, J. Phys. Chem. A 2017, 121, 25, 4754–4764. These results confirm that carbon nitride thin film can catalyze this reaction without the need for an additional co-catalyst.

Q8: There are too many speculations and unsupported claims in the manuscript.

We would like to thank the reviewer for the remark and for the important questions raised in this revision. However, we believe that thanks to the Reviewers' comments in this revision round we have provided significant additional supporting data that are consistent with our findings and claims, including further proofs of the mechanism (TOF, EPR) change in the surface electronic properties by means of changing the solvents, depth profiling analysis, water desorption tests, and more. We hope that the additional set of data provides now solid and convincing arguments for our findings.

Reviewer #3:

Q1: Deconvolution parameters should be given. What kind of constraints were used? Why the FWHM changes only for N-C3 after adsorption of water? It is also observed for C1s components.

We would like to thank the reviewer for this valuable comment. We have updated the tables to include detailed information about the fitting parameters, including the constraints used during deconvolution as it is presented below.

Table S1. N1s XPS fitting parameters. Bare refers to the sample before exposure to heavy water and light illumination in the XPS chamber. Dark and light indicate the absence of illumination from the solar simulator. 0.2, 1, and 2 mbar indicate the heavy water pressure in the characterization chamber. Cycle 1 and cycle 2 indicate subsequent steps of absence and presence of illumination by means of the solar simulator.

Sample	Component	Binding energy (eV)	Line shape	FWHM (eV)	FWHM constrains	Peak area
bare	C-N=C	398.9	LA(1.53,243)	1.5	1.3, 1.551	563003.8
	N-C ₃	400.0		1.5		115541.2
	C-NH _x	401.2		1.5		109912.8
0.2 mbar D ₂ O dark	C-N=C	399.8		1.4		225602.4
	N-C ₃	401.0		1.3		46197.1

	C-NH _x	401.9		1.5		41577.3
0.2 mbar D ₂ O light	C-N=C	399.6		1.4		222548.6
	N-C ₃	400.8		1.2		44509.7
	C-NH _x	401.8		1.4		40058.7
0.2 mbar D ₂ O dark 2 nd cycle	C-N=C	399.8		1.4		207510.1
	N-C ₃	401.0		1.3		43577.1
	C-NH _x	402.0		1.5		37351.8
0.2 mbar D ₂ O light 2 nd cycle	C-N=C	399.6		1.4		215829.1
	N-C ₃	400.8		1.3		43165.8
	C-NH _x	401.8		1.4		38849.2
0.2 mbar D ₂ O KE 400 eV	C-N=C	399.4		1.5		126109.2
	N-C ₃	400.2		1.5		48069.3
	C-NH _x	401.5		1.5		29754.9
0.2 mbar D ₂ O KE 800 eV	C-N=C	399.4		1.5		24173.0
	N-C ₃	400.6		1.5		4859.7
	C-NH _x	401.8		1.5		3853.3
1 mbar D ₂ O dark	C-N=C	399.8		1.6		58405.9
	N-C ₃	401.1		1.2		11681.2
	C-NH _x	401.9		1.5		10513.1
1 mbar D ₂ O light	C-N=C	399.5		1.4		91092.2
	N-C ₃	400.8		1.3		18218.5
	C-NH _x	401.7		1.4		16396.6
2 mbar D ₂ O dark	C-N=C	399.6		1.5		12864.2
	N-C ₃	401.0		1.3		2572.8
	C-NH _x	401.8		1.4		2315.6
2 mbar D ₂ O light	C-N=C	399.6		1.4		14516.7
	N-C ₃	401.0		1.3		2903.3
	C-NH _x	401.8		1.4		2613.0

Table S2. C1s XPS fitting parameters. Bare refers to the sample before exposure to heavy water and light illumination in the XPS chamber. Dark and light indicate the absence and presence of illumination from the solar simulator. 0.2, 1, and 2 mbar indicate the heavy water pressure in the characterization chamber. Cycle 1 and cycle 2 indicate subsequent steps of absence and presence of illumination by means of the solar simulator.

Sample	Component	Binding energy (eV)	Line shape	FWMH (eV)	FWHM constrains	Peak area
bare	C-N=C	288.4	LA(1.53,243)	1.6		881567.1

	C-C	285.2		1.6	1.151, 1.581	1341356.0
	C-NH _x	286.8		1.6		427551.9
0.2 mbar D ₂ O dark	C-N=C	289.3		1.6		487429.6
	C-C	285.9		1.4		2170707.7
0.2 mbar D ₂ O light	C-NH _x	287.3		1.6		259953.1
	C-N=C	289.2		1.5		458086.6
0.2 mbar D ₂ O light	C-C	285.8		1.3		2276500.7
	C-NH _x	287.2		1.5		229043.3
0.2 mbar D ₂ O dark	C-N=C	289.3		1.55		420619.1
	C-C	285.9		1.33		2268165.1
2 nd cycle	C-NH _x	287.3		1.56		210306.6
	C-N=C	289.2		1.50		430202.0
0.2 mbar D ₂ O light	C-C	285.8		1.33		2244084.1
	C-NH _x	287.2		1.55		215101.0
0.2 mbar D ₂ O KE	C-N=C	289.1		1.6		227494.6
	C-C	285.6		1.6		715868.6
400 eV	C-NH _x	287.5		1.6		92407.2
	C-N=C	288.9		1.5		27693.2
0.2 mbar D ₂ O KE	C-C	285.6		1.5		47200.2
	C-NH _x	287.1		1.6		7116.4
1 mbar D ₂ O dark	C-N=C	289.0		1.6		122130.8
	C-C	286.1		1.5		986057.7
1 mbar D ₂ O light	C-NH _x	287.6		1.2		67171.9
	C-N=C	289.1		1.6		207186.1
1 mbar D ₂ O light	C-C	285.7		1.3		744513.16
	C-NH _x	287.1		1.4		103593.1
2 mbar D ₂ O dark	C-N=C	289.0		1.5		39496.7
	C-C	285.7		1.3		212744.6
2 mbar D ₂ O dark	C-NH _x	287.2		1.2		19748.3
	C-N=C	288.9		1.5		45124.2
2 mbar D ₂ O light	C-C	285.7		1.3		230584.9
	C-NH _x	287.2		1.2		21659.6

Table S3. O1s XPS fitting parameters. Bare refers to the sample before exposure to heavy water and light illumination in the XPS chamber. Dark and light indicate the absence and presence of illumination from the solar simulator. 0.2, 1, and 2 mbar indicate the heavy water pressure in the characterization chamber. Cycle 1 and cycle 2 indicate subsequent steps of absence and presence of illumination by means of the solar simulator.

Sample	Component	Binding energy (eV)	Line shape	FWMH (eV)	FWHM constrains	Peak area
bare	C=O	531.6	LA(1.53,243)	2.0	0.53, 13	28832.4
	adventitious oxygen	533.0		2.5		46384.8
0.2 mbar D ₂ O dark	Adsorbed water	533.3		2.2		171941.8
	Water vapor	536.0		0.6		13651.7
0.2 mbar D ₂ O light	Adsorbed water	533.2		2.1		141676.9
	Water vapor	536.0		0.6		13069.5
0.2 mbar D ₂ O dark 2 nd cycle	Adsorbed water	533.3		2.1		146183.5
	Water vapor	536.0		0.6		13654.4
0.2 mbar D ₂ O light 2 nd cycle	Adsorbed water	533.2		2.1		128256.1
	Water vapor	536.0		0.6		12951.4
1 mbar D ₂ O dark	Adsorbed water	533.2		2.2		58423.8
	Water vapor	535.7		0.7		77400.9
1 mbar D ₂ O light	Adsorbed water	533.2		2.0		73169.4
	Water vapor	536.0		0.6		57448.2
2 mbar D ₂ O dark	Adsorbed water	533.3		1.8		19988.03
	Water vapor	536.0		0.5		32444.5
2 mbar D ₂ O light	Adsorbed water	533.3		1.9		22967.8
	Water vapor	535.9		0.6		32755.1

Table S4. Au4f XPS fitting parameters. Bare refers to the sample before exposure to heavy water and light illumination in the XPS chamber. Dark and light indicate the absence and presence of illumination from the solar simulator.

Sample	Component	Binding energy (eV)	Line shape	FWMH (eV)	FWHM constrains	Peak area
bare	Au4f _{5/2}	87.6	DS(0.01,250)	0.6	0.13, 3.25	2126675.2
	Au4f _{7/2}	83.9		0.6		2872801.0

0.2 mbar	Au4f _{5/2}	87.6		0.6		1948930.6
D ₂ O dark	Au4f _{7/2}	83.9		0.6		2717946.5
0.2 mbar	Au4f _{5/2}	87.6		0.6		1843985.3
D ₂ O light	Au4f _{7/2}	83.9		0.6		2515657.1

Table S5. N1s XPS fitting parameters. Bare refers to the sample before exposure to methanol in the XPS chamber. Dark indicates the absence of illumination from the solar simulator. 0.2, indicates the methanol pressure in the characterization chamber. Desorption indicates subsequent step of methanol desorption by means of chamber evacuation.

Sample	Component	Binding energy (eV)	Line shape	FWHM (eV)	FWHM constrains	Peak area
bare	C-N=C	399.6	LA(1.53,243)	1.6	1.3, 1.551	299936.5
	N-C ₃	400.8		1.4		59987.3
	C-NH _x	402.0		1.4		53988.6
0.2 mbar methanol dark	C-N=C	400.0		1.6		164935.0
	N-C ₃	401.2		1.3		32987.0
	C-NH _x	402.1		1.5		29688.3

Table S6. N1s XPS fitting parameters. Bare refers to the sample before exposure to isopropanol in the XPS chamber. Dark indicates the absence of illumination from the solar simulator. 0.2, indicates the isopropanol pressure in the characterization chamber. Desorption indicates subsequent step of isopropanol desorption by means of chamber evacuation.

Sample	Component	Binding energy (eV)	Line shape	FWHM (eV)	FWHM constrains	Peak area
bare	C-N=C	399.6	LA(1.53,243)	1.5	1.3, 1.551	191476.1
	N-C ₃	400.8		1.4		47869.0
	C-NH _x	401.9		1.5		38295.2
0.2 mbar isopropanol dark	C-N=C	399.6		1.6		42369.5
	N-C ₃	400.9		1.3		11863.5
	C-NH _x	401.8		1.6		9321.3

Table S7. N1s XPS fitting parameters. Bare refers to the sample before exposure to toluene in the XPS chamber. Dark indicates the absence of illumination from the solar simulator. 0.2, indicates the toluene pressure in the characterization chamber. Desorption indicates subsequent step of toluene desorption by means of chamber evacuation.

Sample	Component	Binding energy (eV)	Line shape	FWHM (eV)	FWHM constrains	Peak area
bare	C-N=C	399.6	LA(1.53,243)	1.6	1.151, 1.551	218116.2
	N-C ₃	400.7		1.4		43623.2
	C-NH _x	401.9		1.3		39260.9
0.2 mbar toluene dark	C-N=C	399.8		1.6		69326.5
	N-C ₃	401.1		1.3		13865.3
	C-NH _x	402.0		1.4		12478.8

Table S8. N1s XPS fitting parameters. Bare refers to the sample before exposure to heavy water in the XPS chamber. light indicates the presence of illumination from the solar simulator. 0.2, indicates the heavy water pressure in the characterization chamber.

Sample	Component	Binding energy (eV)	Line shape	FWHM (eV)	FWHM constrains	Peak area
0.2 mbar D ₂ O light irradiation 0 min	C-N=C	400.0	LA(1.53,243)	1.6	1.151, 1.551	42943.8
	N-C ₃	401.3		1.2		9447.6
	C-NH _x	402.1		1.4		8588.8
0.2 mbar D ₂ O light irradiation 75 min	C-N=C	400.0		1.6		74874.2
	N-C ₃	401.2		1.2		14974.8
	C-NH _x	402.1		1.4		13477.4
0.2 mbar D ₂ O light irradiation 120 min	C-N=C	400.0		1.6		75638.1
	N-C ₃	401.2		1.2		15127.6
	C-NH _x	403.7		1.6		13614.9

Table S9. C1s XPS fitting parameters. Bare refers to the sample before exposure to methanol in the XPS chamber. Dark indicates the absence of illumination from the solar simulator. 0.2, indicates the methanol pressure in the characterization chamber. Desorption indicates subsequent step of methanol desorption by means of chamber evacuation.

Sample	Component	Binding energy (eV)	Line shape	FWHM (eV)	FWHM constrains	Peak area
bare	C-N=C	289.4	LA(1.53,243)	1.6	1.151,	475788.8
	C-C	286.1		1.6	1.581	592222.1

	C-NH _x	287.7		1.6		130333.2
0.2 mbar methanol dark	C-N=C	289.4		1.6		291517.6
	C-C	286.2		1.5		388366.5
	C-NH _x	287.8		1.5		145443.8

Table S10. C1s XPS fitting parameters. Bare refers to the sample before exposure to isopropanol in the XPS chamber. Dark indicates the absence of illumination from the solar simulator. 0.2, indicates the isopropanol pressure in the characterization chamber. Desorption indicates subsequent step of isopropanol desorption by means of chamber evacuation.

Sample	Component	Binding energy (eV)	Line shape	FWHM (eV)	FWHM constrains	Peak area
bare	C-N=C	289.0	LA(1.53,243)	1.6	1.151, 1.551	247483.5
	C-C	285.7		1.5		1021754.8
	C-NH _x	287.3		1.6		110865.6
0.2 mbar isopropanol dark	C-N=C	289.0		1.6		107849.8
	C-C	285.8		1.4		594385.2
	C-NH _x	287.2		1.6		129924.8

Table S11. C1s XPS fitting parameters. Bare refers to the sample before exposure to toluene in the XPS chamber. Dark indicates the absence of illumination from the solar simulator. 0.2, indicates the toluene pressure in the characterization chamber. Desorption indicates subsequent step of toluene desorption by means of chamber evacuation.

Sample	Component	Binding energy (eV)	Line shape	FWHM (eV)	FWHM constrains	Peak area
bare	C-N=C	289.4	LA(1.53,243)	1.6	1.151, 1.551	313328.1
	C-C	285.9		1.6		476962.8
	C-NH _x	287.6		1.6		82340.0
0.2 mbar toluene dark	C-N=C	289.2		1.6		132692.5
	C-C	285.9		1.5		277736.2
	C-NH _x	287.5		1.6		48959.5

Table S12. N1s XPS fitting parameters. Bare refers to the sample before exposure to heavy water in the XPS chamber. light indicates the presence of illumination from the solar simulator. 0.2, indicates the heavy water pressure in the characterization chamber.

Sample	Component	Binding energy (eV)	Line shape	FWHM (eV)	FWHM constrains	Peak area
0.2 mbar D ₂ O light irradiation 0 min	C-N=C	289.2	LA(1.53,243)	1.6	1.151, 1.551	139911.5
	C-C	286.1		1.5		836383.8
	C-NH _x	287.7		1.2		69955.8
0.2 mbar D ₂ O light irradiation 75 min	C-N=C	289.3		1.6		161098.9
	C-C	286.0		1.5		623449.7
	C-NH _x	287.5		1.5		80549.5
0.2 mbar D ₂ O light irradiation 120 min	C-N=C	289.3		1.5		175192.4
	C-C	286.0		1.5		538010.6
	C-NH _x	287.5		1.6		87596.2

Q2: How can the shift of XPS peaks to higher binding energies after D₂O adsorption be explained? Is it solely due to a charging effect or to a real modification of the atoms' chemical environment?

We appreciate the reviewer's valuable comment. The adsorption of heavy water molecules on the carbon nitride causes a shift in the maxima of the XPS peaks, as shown in Figure R10. We attribute this effect to the formation of hydrogen bonds between the heavy water and the carbon nitride, where the N atoms of the carbon nitride act as hydrogen bond donors, donating electron density to the heavy water (acceptor). As we mentioned in the manuscript, this interaction reduces the electron density on the semiconductor's surface layer, which is also reflected in the shift of carbon's spectroscopic features toward higher binding energies (Figure R10 B). The shift is particularly pronounced at the catalyst's edge functionalities (-C-N=C-), indicating preferential adsorption at these sites, as further supported by the NEXAFS spectra (Figures 1C and 1D). Furthermore, previously reported theoretical calculation have demonstrated a strong adsorption of the water at the carbon nitride functionalities (e.g. *Phys. Chem. Chem. Phys.*, 16, 15917-15926 (2014), *J. Phys. Chem. Lett.*, 9(16), 4695-4699 (2018), *Phys. Chem. Chem. Phys.*, 16, 3299-3304 (2014), *J. Phys. Chem. A*, 121(25), 4754-4764 (2017)). In particular Wu et al. (*Phys. Chem. Chem. Phys.*, 16, 3299-3304 (2014)) have shown in their theoretical study that the adsorption of water on the carbon nitride causes a buckling of the heptazine unit and a persistent change of the electronic properties of the material, accounting for a shift of the VB values up to +2.5 eV. Although here we cannot confirm experimentally the "buckling" of the structure, the effect on the electronic properties is in good agreement with our experimental findings. We convey that the adsorption of water causes a redistribution of the electronic density at the carbon nitride atoms, with a potential change of configuration of the heptazine unit without any chemical reaction occurring.

Figure R10. Effect of adsorption of heavy water on the carbon nitride thin film. (A) XPS, N1s spectra of carbon nitride thin film bare (upper panel) and after adsorption of heavy water (lower panel); arrows indicate spectral shift direction. (B) XPS, C1s spectra of carbon nitride thin film bare (upper panel) and after adsorption of heavy water (lower panel); arrows indicate spectral shift direction. (C) NEXAFS, N K-edge of the carbon nitride thin film bare (black line) and after adsorption of heavy water (magenta line); colored boxes highlight the different chemical features and the arrows indicate the spectral changes intensity. (D) NEXAFS, C K-edge of the carbon nitride thin film bare (black line) and after adsorption of heavy water (magenta line); colored boxes highlight the different chemical features and the indicate the spectral changes intensity. (E) VB-XPS of the carbon nitride thin film bare (black line) and after adsorption of heavy water (magenta line) with relative valence band values, arrow indicates

spectral change direction. (F) Schematic representation of the carbon nitride-heavy water structure after adsorption.

Q3: Authors stated that the hole created by the photochemical excitation is transferred to the heavy water, forming a surface $\text{DO}\cdot$ radical. But there is no proofs for that. Figure 2F shows only a schematic representation. The formation of these radicals could be observed in other experimental measurements such as EPR/lights.

The authors thank the reviewer for raising this important point that enables us to depict more convincingly the photocatalytic mechanism. To confirm the formation of radicals we performed the EPR experiment with radical trap 5,5-dimethyl-1-pyrroline N-oxide (DMPO). The EPR spectrum with discussion of obtained results is shown in Supplementary note 4 and below.

Figure R11. EPR spectra of reaction mixture with DMPO during light illumination (upper panel) and in the dark.

The presence of multiple peaks in the EPR spectrum under 415 nm light suggests the generation of radical species during the photocatalytic reaction. The spectrum displays four main lines with a 1:2:2:1 intensity ratio, characteristic of a DMPO-OH adduct, indicating the presence of hydroxyl radicals ($\cdot\text{OH}$), which result from trapping these radicals during the photocatalytic process (g factor 2.006). While the DMPO-OH adduct is prominent, there are also contributions from superoxide anions ($\text{O}_2^{\cdot-}$), trapped as a DMPO-OOH adduct (g factor 2.004), and from a DMPO-alkyl adduct (g factor 2.006). The absence of the sharp, narrow peaks typically associated with DMPO-alkyl adducts suggests that alkyl radicals may not be the dominant species, although their contribution cannot be entirely ruled out (Phys. Chem. Chem. Phys. 16, 15917-15926 (2014)).

The EPR spectrum recorded in the dark shows no significant peaks, confirming that the radical species observed under illumination are specifically generated by the photocatalytic activity of the carbon nitride thin film.

Q4: Concerning the mechanism. What is the exact mechanism of formation of the N-photoelectron $\cdot\cdot\cdot$ D⁺ and N $\cdot\cdot\cdot$ D-O $\cdot\cdot\cdot$ C species? How can their existence be confirmed by other experimental techniques?

In our study, we propose a Proton-Coupled Electron Transfer (PCET) mechanism to explain the reaction pathway. Upon photoexcitation, electrons from the nitrogen site in carbon nitride are excited and transiently interact with nearby deuterium ions, formed through the partial dissociation of heavy water. This interaction stabilizes the excited electron on the deuterium, resulting in the formation of the N-photoelectron $\cdot\cdot\cdot$ D⁺ species. Simultaneously, deprotonated oxygen from the water molecule can interact with another nitrogen site, potentially transferring or sharing electron density with adjacent carbon atoms. This interaction leads to the formation of the radical species N $\cdot\cdot\cdot$ D-O $\cdot\cdot\cdot$ C, which may facilitate further catalytic activity on the surface. Consequently, we have applied and were granted another beamtime we investigated the stability of the carbon nitride film under extended illumination and monitored additional products formation with the TOF-MS detector.

Initially, we observed a gradual decrease in the signal associated with D₂O, accompanied by corresponding increases in the TOF-MS signals for $\cdot\cdot\cdot$ OOD, D₂O₂, O₂, and D₂. After an initial stabilization, the system reaches a steady state, with a constant evolution of the products and negligible differences in the TOF-MS signals over the time of the experiment (about 2 hours). It is important to note that the signal for D₂O₂ was recorded and consistently present during the photocatalytic reaction, supporting the mechanistic hypothesis of a proton-coupled electron transfer mechanism (PCET), with the formation of D₂O₂ as an intermediate reaction product.

Figure R12. TOF-MS signal of OOD, D₂O₂, O₂, D₂ and D₂O; grey areas indicate the absence of illumination; yellow areas indicate the time the carbon nitride film was subjected to illumination by means of the solar simulator.

Moreover, to confirm the presence of these species and support the proposed PCET mechanism, we employed EPR spectroscopy (see Supplementary Note 4 and Reviewer #3 Q3). This technique is crucial for detecting the unpaired electrons characteristic of radical species like N \cdots D-O \cdots C, providing direct evidence of their formation and stability.

Q5: How does the catalyst's stability evolve over time under light irradiation? Are there any signs of deactivation or modification of the material's structure?

We would like to thank the reviewer for this important question pointing to the long term stability of our catalyst. Based on our XPS and VB-XPS analyses, we did not observe any significant changes in the overall structure of the carbon nitride photocatalyst after prolonged light exposure, indicating that the material remains stable during photocatalysis. However, we did note a

significant change in the population of the features in the C1s and N1s spectrum with longer illumination time. This behavior is discussed in detail in Supplementary Note 3 and below.

Light stability is a critical factor in the performance of photocatalysts, as it determines their ability to maintain long-term photocatalytic activity under continuous light exposure. To assess this, we monitored the N1s, C1s, and valence band spectral features of carbon nitride to evaluate the stability of its electronic structure during prolonged light illumination under photocatalytic conditions (Figure R13).

The N1s and C1s XPS spectra indicate that the overall structure of carbon nitride remains mostly unchanged after 2 hours of light illumination. This suggests that the carbon nitride framework is stable during photocatalysis, which is crucial for maintaining its long-term activity as a photocatalyst. However, a more detailed analysis of the C1s spectrum reveals a change in the population of the C-N=C feature with increasing illumination time up to 20%. We attribute this behavior to the desorption of surface contaminants, specifically adventitious carbon species, under prolonged illumination. The VB-XPS spectra do not show significant changes, supporting the hypothesis that the surface electronic structure of the carbon nitride remains stable during the photocatalytic process. This aligns with the stability observed in the core-level XPS spectra and the desorption of adventitious carbon species, which do not contribute to the electronic properties change on the carbon nitride film surface. Furthermore, as shown in Q4 of this reviewer we record a constant evolution of the products over about 10000 seconds (ca. 167 minutes) with no significant changes over time.

Figure R13. XPS spectra of N1s (A) and C1s (B) and VB-XPS (C) of carbon nitride under 0.2 mbar of D₂O at the beginning of light illumination (top panel), after 75 min of illumination (middle panel) and after 120 min of illumination (bottom panel).

Q6: What other semiconductor materials could exhibit similar behaviors? To confirm these findings.

We thank the reviewer for the insightful comment that gives broader perspective on our work. From the perspective of the change of electronic properties upon water adsorption in a covalent

system, we believe that semiconductor carbon organic frameworks (COF) with heterocycle nitrogen functionalities may exhibit similar behavior. Potentially a further perspective will involve the orientation of the N's p orbital that can play a fundamental role in the interaction with the water and the resulting change of the electronic properties in the semiconductor. Additionally, metal organic frameworks could show analogous behavior when they possess functionalities able to act as a donor for the formation of hydrogen bonds with water. Examples can be found within MOFs and COFs structures, which have been demonstrated to possess photocatalytic activity in water splitting (*ChemSusChem*, 16, e202300021 (2023); *International Journal of Hydrogen Energy*, 51, 376-398, (2024)). On the other hand, the activity of well-known metal oxide photocatalysts, such as hematite ($\alpha\text{-Fe}_2\text{O}_3$) (*The Journal of Physical Chemistry Letters* 7(7), 1155-1160 (2016)) and titania (*The Journal of Physical Chemistry B* 110 (37), 18492-18495 (2006)), depending on the surface termination and functional groups can act as hydrogen bond donor or acceptor with water. We would like to point to the critical role of the surface functionalities for the formation of hydrogen bonds and therefore the displacement of the electron density at the photocatalyst.

By conducting parallel studies with these semiconductors, we can determine whether the findings observed with carbon nitride are unique or if they share common photocatalytic mechanisms. Such comparisons would provide valuable insights into the broader applicability of the photocatalytic processes identified in our study and may open the door to new strategies for enhancing water splitting efficiency across various materials.

Reviewers' comments in black – Authors' responses in blue

Reviewer #1 (Remarks to the Author):

I believe the authors have adequately addressed my comments, and I recommend this manuscript for acceptance.

The authors thank this Reviewer for the acknowledgment, the recommendation, and the comments that enabled us to improve the quality of our manuscript.

Reviewer #2 (Remarks to the Author):

The authors addressed the remarks.

The authors thank this Reviewer for the acknowledgment and the comments that enabled us to improve the quality of our manuscript.

Reviewer #3 (Remarks to the Author):

The authors have replied to all referee questions and remarks. They have improved their manuscript. I recommend now the article for publication.

The authors thank this Reviewer for the acknowledgment, the recommendation, and the comments that enabled us to improve the quality of our manuscript.